# A Comprehensive Review of COVID-19-Infection- and Steroid-Treatment-Associated Bone Avascular Necrosis: A Multi-Study Analysis

**DOI:** 10.3390/diagnostics14030247

**Published:** 2024-01-24

**Authors:** Evangelos Sakellariou, Evangelia Argyropoulou, Panagiotis Karampinas, Athanasios Galanis, Iordanis Varsamos, Vasileios Giannatos, Elias Vasiliadis, Angelos Kaspiris, John Vlamis, Spiros Pneumaticos

**Affiliations:** 1Department of Orthopaedic Surgery, National & Kapodistrian University of Athens, KAT General Hospital, 14561 Athens, Greece; vagossak@hotmail.com (E.S.); karapana@yahoo.com (P.K.); athanasiosgalanis@yahoo.com (A.G.); jordan.var1995@gmail.com (I.V.); eliasvasiliadis@yahoo.gr (E.V.); angkaspiris@hotmail.com (A.K.); jvlamis@email.com (J.V.); spirospneumaticos@gmail.com (S.P.); 2Department of Orthopaedics and Traumatology, University General Hospital of Patras, 26504 Patras, Greece; vasileiosgiannatos@outlook.com

**Keywords:** COVID-19, AVN, bone avascular necrosis, osteonecrosis, hip pain, steroids

## Abstract

Background: The COVID-19 pandemic has presented numerous challenges to the global healthcare system and emerging evidence suggests a potential link between COVID-19 treatment, specifically steroid therapy, and the development of avascular necrosis (AVN) of the hip. This review aims to provide a comprehensive overview of recent studies and case reports investigating the relationship between COVID-19, corticosteroid therapy, and the development of AVN. Understanding the nuances of AVN in the context of COVID-19 is crucial for healthcare professionals to navigate treatment decisions and mitigate potential complications. Materials and Methods: The Preferred Reporting Items for Systematic Reviews and a Meta-Analysis (PRISMA) were used in the systematic review until 1 September 2023. The full texts of the remaining twenty-three (*n* = 23) articles were independently reviewed by the authors of this review. Conclusions: The association between steroid therapy for COVID-19 and the development of hip AVN is a noteworthy concern even though no relationship is evident between the duration of treatment, cumulative dosage of medication, maximum daily dosage received, and presence of AVN. Further research with larger cohorts and long-term follow up is needed to better understand the causative relationship and optimal management strategies for hip AVN in the context of COVID-19 and steroid therapy.

## 1. Introduction

The COVID-19 pandemic has presented numerous challenges to the global healthcare system, and as healthcare professionals continue to explore the long-term consequences of the infection, emerging evidence suggests a potential link between COVID-19 infection and treatment, specifically steroid therapy, and the development of avascular necrosis (AVN) of the hip. 

COVID-19 infection is tethered to endothelial damage and with the combination of steroid therapy, microthrombi begin to form and can cause osteonecrosis. COVID-19 triggers systemic inflammation, leading to an influx of cytokines that prevent the osteoblast from proliferating and differentiating. Moreover, pro-inflammatory proteins cause vasculitis, hypercoagulability, and bone necrosis, and the result can be avascular necrosis (AVN) [1]. More specifically, a state of disrupted molecular pathways exists that consists of endothelial dysfunction, coagulopathy, and impaired angiogenesis, which lead to the initiation and progression of bone ischemia [2]. Concerning the endothelial dysfunction, the balance between vasodilation and vasoconstriction is affected, causing ischemia and disrupted blood flow and thrombi. As for the coagulopathy, it leads to the flustered relation of plasminogen activator inhibition and the tissue plasminogen activator reducing the fibrinolysis and increasing the clot formation [3]. Moreover, the overexpressed angiogenesis can cause unusual bone remodeling. While various factors contribute to AVN, emerging evidence suggests a potential association between osteonecrosis and COVID-19, the disease originated by the novel coronavirus. The interplay between viral infections, the immune response, and the use of corticosteroids, a common therapeutic intervention for severe COVID-19 cases, has raised concerns about its impact on joint health.

AVN is a debilitating condition characterized by the death of bone tissue due to a lack of blood supply, often leading to joint destruction and significant morbidity. Studies report 12% of joint pain following the infection and 9% of patients complain about mobility difficulties [4]. AVN can be found mainly after prolonged therapies with a high dose of steroid treatment, probably due to a combination of fat embolism, lipid abnormalities, and coagulation defects [5]. The altered lipid metabolism can result in fat accumulation in bone marrow, which will increase the intramedullary pressure and decrease the blood supply of the bones. The estimated cumulative dose, which was measured at 2000 mg of prednisolone, is a risk of osteonecrosis (ONC) and depends on the type of steroid, the daily dosage, and the duration of treatment [6]. In addition to this, the COVID-19 antiviral treatments, like ritonavir and lopinavir, are incriminated for ONC [7].

The incidence of hip AVN in patients recovering from COVID-19 and receiving steroid therapy varies among studies and there is no report until now establishing the incidence of femoral head osteonecrosis (FHOn). Even though the exact cumulative steroid dose and the duration of COVID-19 treatment that can lead to AVN are not yet established, it is known that a longer treatment duration, the severity of infection, admission in the ICU, and the personal history count as potential risk factors [8]. Close follow up with MRI of the major joints is suggested for the first 3 months after a SARS infection, in order to treat the damage and save the joint, especially in young active patients (Figure 1). Symptoms of hip AVN typically manifested several weeks to months after COVID-19 infection and steroid treatment. The literature illustrates the importance of MRI in the diagnosis of AVN, and more specifically as an early-stage diagnosis [9], as it may aid in choosing between a surgical or nonsurgical type of treatment [10]. The femoral head is the most common location for ONC and is quite often bilateral [11], but signs of necrosis can be found in other parts like the shoulder, knee, and ankle [12]. There appears to be a correlation between the use of steroids, particularly methylprednisolone and dexamethasone, in the treatment of moderate to severe COVID-19 and the subsequent development of hip AVN [13]. Further research with larger cohorts and long-term follow ups is needed, to better understand the causative relationship and optimal management strategies for hip AVN in the context of COVID-19 and steroid therapy.

This review aims to provide a comprehensive overview of recent studies and case reports, investigating the relationship between COVID-19, corticosteroid therapy, and the development of AVN. The presented cases span diverse demographics and severity levels, shedding light on the clinical outcomes and management strategies employed. Understanding the nuances of AVN in the context of COVID-19 is crucial for healthcare professionals to navigate treatment decisions and mitigate potential complications.

## 2. Materials and Methods

The Preferred Reporting Items for Systematic Reviews and a Meta-Analysis (PRISMA) were used in the systematic review and no ethical approval was needed, because the information was from published literature. The studies that were identified and included were written in the English language and designed to illustrate the appearance of bone avascular necrosis after COVID-19 infection or vaccination. A comprehensive and systematic search for articles published in computerized literature databases (MEDLINE/Pubmed and Google Scholar) from 2022 to September 2023, containing the keyword terms COVID-19 and joint pain and bone avascular necrosis and bone osteonecrosis, was performed and illustrated at the PRISMA flow diagram (Figure 2). Reference lists from articles that met the inclusion criteria were further investigated.

Relevant studies were selected for inclusion based on the following predetermined eligibility criteria:Articles in EnglishCOVID-19 infection or vaccinationNo joint pain prior to the infectionMusculoskeletal symptoms after COVID-19Patients over 18 years old

The search using the aforementioned keywords yielded 13,654 articles, until 1 September 2023. The studies were analyzed for duplication with the resulting number of studies becoming 102. The authors independently reviewed the titles and abstracts of each result, and those that were clearly irrelevant and/or failed to pertain to the pre-determined inclusion criteria (*n* = 39) were eliminated. The remaining 75 (*n* = 30) articles were further scrutinized for clearly relevant trials that indisputably met the inclusion criteria, eliminating a further 7 trials. The full texts of the remaining 23 (*n* = 23) articles were independently reviewed by the authors of this review, who agreed upon all 23 remaining articles to be objectively relevant to this summary in discussion (Table 1). 

## 3. Results

### 3.1. Demographics

Twenty-three articles are included in the study and there are 613 patients in total. In total, 62% of them are males and the rest are females. The estimated mean age is 42.9 years old, ranging from 19 to 63 years old. It is important to note that a lot of patients had bilateral joint pain and signs of AVN after the COVID-19 infection (Figure 1). The majority of the articles are case reports, illustrating mostly 1 patient and some of them reference up to 5 different patients [14]. The prospective study by Veizi et al. [9] is the biggest study so far, concerning the number of patients, with 472 people. All of the patients showed symptoms of hip osteonecrosis, except for the study by Kashkosh et al. [15], in which the patient had humeral head pain, only two days after the second dose of the Pfizer COVID-19 vaccine (Figure 3 and Figure 4).

### 3.2. Course of COVID-19 Infection 

The infection’s severity varied, with mild, moderate, and severe cases being observed. Mild infections in the studies were scarce with only 11 patients, while the majority of the patients had mostly moderate and some moderate to severe symptoms. In total, 18 out of 23 articles are related to severe COVID-19 infection.

Clinical signs of musculoskeletal symptoms typically manifested within a range of days to a few months after the COVID-19 infection, with an average of 85 days. The shortest period is 14 days [16] and the longer duration for the AVN symptoms to appear is 300 days [17]. Generally, joint discomfort occurred after the resolution of acute respiratory symptoms and elevated body temperature.

### 3.3. COVID-19 Treatment

The treatment for moderate to severe cases consists of antiviral therapy and corticosteroids, which could be administered either intravenously or orally. Three studies combined the antiviral medication with steroids [18,19,20], while at the same time, three other studies did not use any corticosteroid [21,22,23]. However, Veizi et al. [9] divided the patients into two groups and half of them were treated with steroids and the rest without. Corticosteroids commonly used in COVID-19 treatment are dexamethasone, methylprednisolone, and prednisolone and are associated with potential complications such as femoral head osteonecrosis (FHOn). Their doses varied widely, ranging from 40 mg to 20,675 mg of prednisolone equivalents, while the treatment duration differed from a few days to several weeks. A cumulative dose of 2000 mg of prednisolone or an equivalent is linked to an increased risk for AVN. While corticosteroids have shown benefits in treating severe COVID-19 cases, the text raises concerns about potential long-term complications. It emphasizes the need for a balance between the benefits of treatment and the risks of complications.

### 3.4. AVN Treatment Approaches and Outcome Measures

In six studies, the patients had initially undergone conservative treatment, which was constituted by NSAIDS, physical therapy [18,22], intra-articular (IA) steroid injections, oral or IV steroid medication, and bisphosphonates [21,24]. Concerning the mobility, patients modified their activities and had protected bare weight. In one study, the patients received an IA hydrodilatation injection [15]. Out of those articles, the conservative treatment succeeded only in two studies, with clinical improvement [18] and a mean VAS score of 2.7 [24], while in the rest, the patients had surgical intervention, like core decompression, total hip arthroplasty (THA), and bone marrow aspirate concentrate (BMAC) injection. As for core decompression, it was combined with BMAC in three studies [19,25,26] out of the ten that it was used in. Furthermore, total hip arthroplasty was paired with the decompression of the necrosis foci in the study by Annam et al. [20] for a younger patient. For the rest of the patients, they were treated with THA, and the results indicated an improvement in motor activity and decrease in pain intensity, without a significant improvement in MRI though (Figure 5). The mean VAS score lessened from 9.4 to 2.8 in the first postoperative week [14]. Some cases did not receive specific treatment for AVN and in general the response to treatment varied, while some patients improved with conservative measures, while others required surgical intervention. Only one case involved infection post-THA and needed a two-stage THA [14]. As for the radiological outcomes, in some cases, there was improvement seen in imaging studies, but not in all. However, Veizi et al. [9] highlighted that there is no relationship between the treatment duration, cumulative dosage of medication, and ONC.

### 3.5. Follow-Up Period

The text highlights persistent symptoms, including fatigue, shortness of breath, anxiety, depression, joint pain, and stiffness, in COVID-19 survivors at a one-year follow up. Joint pain and stiffness are underreported in long-term studies. The majority of the studies had followed up with their patients for an average of 3–4 months and only one study had long-term outcomes at 25 months [27]. The impact of SARS-CoV-2 infection and therapeutic interventions on the skeletal system, especially AVN, is not thoroughly investigated. The text emphasizes the need for further studies to identify risk factors, determine the incidence of FHOn, and establish protocols for an early diagnosis and intervention. The study also highlights gaps in existing studies, such as inconsistent reporting of musculoskeletal symptoms and a lack of emphasis on FHOn incidence. It suggests that future research should include comprehensive musculoskeletal evaluations in the post-acute phase of COVID-19.

**Table 1 diagnostics-14-00247-t001:** Characteristics of the included studies.

Scheme	Study Design	No. of Patients and Gender	Mean Age (years)	COVID-19 Symptoms	Treatment of COVID-19	Days Until AVN	AVN Treatment	Results
1. Sulewski et al. (2021) [5]	Cohort study	10 (6 F + 4 M)	58.8	Moderately severe	Steroids		Initially conservatively (NSAIDs, IA steroid inj) with no improvement Oral dexamethasone at 2 × 8 mg daily for 2 weeks	30% THA with good clinical outcome 10% chronic pain without joint destruction in the control tests
2. Alkindi et al. (2021) [18]	Case report	1 M	29		Combination of experimental anti-COVID-19 therapies (favipiravir, hydroxychloroquine, tocilizumab) Methylprednisolone IV at 40 mg/d for 5 days		IV corticosteroids for 5 days (cumulative dose: 200 mg)	Clinical improvement with NSAIDs and PT
3. Daltro et al. (2021) [28]	Follow up	14 M + 9 F	43.5	33% Mild 66% Moderate/Severe	Mild infection: no hospitalization/corticosteroids Dexamethasone: min dose of 40 mg/day for a mean time of 14.6 days (min 15–max 21)			33% osteonecrosis of the femoral head
4. Chacko et al. (2021) [29]	Case report	1 M (bilateral)	23		IV dexamethasone at 6 mg/d for 9 days and IV methylprednisolone at 40 mg × 2/d for 5 days Cumulative dose is equivalent to 860 mg of prednisolone	56	Core decompression of femoral heads Bone marrow aspirate concentrate injection	
5. Joshi et al. (2021) [30]	Case report	1 F (bilateral)	31	Moderate/Severe	Methylprednisolone at 32 mg/d for 7 days with minimal improvement Continued for 10 days more Total dose of 544 mg	30	N/G	
6. Agarwala et al. (2021) [24]	Case report	3 M (bilateral)	37	Moderate/Severe	Mean equivalent to 758 mg of prednisolone	58	Oral alendronate at 70 mg/w IV zoledronic acid at 5 mg annually	Mean VAS: 2.7 No surgery
7. Panin et al. (2022) [19]	Case report	4 (2 M/2 F) (bilateral)	34	Moderate/Severe	Mean total dose of dexamethasone/prednisolone of 264 (80–600 mg)/1759 (533–4000 mg) 1 patient: iv favipiravir, tocilizumab 2nd: iv triazaverin	96.6	2 decompressions of the necrosis foci Administration of a bone marrow concentrate 1THA Statins Bisphosphonates Anticoagulants	Improvement in motor activity Decrease in pain intensity No significant improvement in MRI
8. Uyshal et al. (2022) [21]	Case report	1 M	63	Mild/Moderate	No steroid usage		Protected weight bearing Oral alendronate at 70 mg/w (no improvement) THA	
9. Ergün et al. (2022) [22]	Case report	1 F (bilateral)	51		No prior use of steroid Favipiravir LMWH	60	Core decompression PT No weight bearing for 6 weeks	Improved clinical scores No femoral head subchondral bone collapse
10. Kingma et al. (2022) [17]	Case report	1 M (bilateral)	60	Severe	Total dose of prednisone equivalent of 1327.5 mg	300	Bilateral THA	No complications
11. Ardakani et al. (2022) [14]	Case series	5 (2 M/3 F)	38.4	Moderate/Severe	Mean dose of prednisolone was 1695.2 mg	41.6	All patients underwent surgery with direct anterior approach 1 did two-stage THA due to Serratia marcescens infection in both hips	Clinical and laboratory symptoms improved significantly Mean VAS decreased from 9.4 to 2.8 1 week post-operation
12. Annam et al. (2022) [20]	Case report	2 M (bilateral)	48 (27/69)	Moderate	Oseltamivir, doxycycline, and methylprednisolone Mean total dose of methylprednisolone of 588 mg, equivalent to 735 mg of prednisolone		The younger had bilateral THA and hip core decompression The older had left THA and right hip decompression	
13. Kamani et al. (2022) [31]	Case report	1 M (bilateral)	40	Severe	Steroid injection		Bilateral core decompression hip surgery PT	
14. Kashkosh et al. (2022) [15]	Case report	1 M	40	Second dose of the Pfizer COVID-19 vaccine	AVN of the humeral head	2	Analgesics Activity modification IA hydrodilatation inj	Improved ROM Severe shoulder pain Surgical intervention
15. Jyothiprasanth et al. (2023) [23]	Cohort study	17 (10 M/7 F)/4 bilateral	37	82.4% COVID-19 Inf	No steroid therapy	66	N/G	
16. Baimukhamedov (2023) [10]	Cohort study	8 M	N/G	N/G	Range of cumulative corticosteroid doses 50–20,675 mg of prednisolone	N/G	N/G	
17. Karpur et al. (2023) [32]	Follow up	20 (14 M/6 F)	N/G	N/G	N/G	N/G	N/G	Stage I: 45% Stage II: 40% M/F: 70/30%
18. Shershah et al. (2023) [25]	Case reports	3 (2 M/1 F) bilateral	29.3	Severe	560 mg of IV methylprednisolone	240	Conservative without results 2 bilateral THAs 1 core decompression with bone marrow aspirate infiltration	
19. Velchov et al. (2023) [27]	Follow up	24 (17 M/7 F) 4 bilateral	55.6	8 Moderate/16 Severe	Moderate: mean of 120 mg of dexamethasone Severe: also 3600 mg of methylprednisolone	56.3	23 THAs 5 core decompressions	
20. Jayapalan et al. (2023) [26]	Case report	1 M (bilateral)	31	Moderate	IV methylprednisolone (600 mg) followed by an oral dose of 8 mg	65	Bilateral core decompression and BMAC	
21. Parikh et al. (2023) [33]	Case reports	3 (2 M/1 F)	55.6	Moderate	Steroid treatment	N/G	Core decompression (1 bilateral)	
22. Sinha et al. (2023) [16]	Cohort study	10 (4 M/6 F)	53.9	7 Moderate/3 Severe	4 steroid therapies	14	4 core decompressions	
23. Veizi et al. (2023) [9]	Prospective study	472 (289 M/183 F)	42		Group 2: (236) received steroid treatment		Increased % of AVN in Group 2 Joint pain: 5.1% in Group 1 11.9% in Group 2 AVN: 8 pts from Group 2	No relationship between the treatment duration, cumulative dosage of medication, and ONC

N/G: Not Given.

## 4. Discussion

As the medical community strives to comprehend the multifaceted implications of COVID-19 and its treatments, this review contributes to the ongoing dialogue surrounding the intricate relationship between steroid therapy and the risk of hip AVN. Understanding these associations is crucial for forming clinical decision making and developing targeted interventions to optimize patient outcomes in the aftermath of COVID-19 infection. This review article aims to provide a comprehensive analysis of avascular necrosis (AVN) cases in the orthopedic department, associated with steroid use in the context of COVID-19 treatment. The included studies present a diverse range of cases, encompassing various patient demographics, severities of COVID-19, steroid regimens, and outcomes. The objective is to synthesize the existing literature, identify common trends, and draw insights into the management and potential preventive strategies for steroid-induced AVN.

Bone avascular necrosis due to steroid treatment existed even before the COVID-19 pandemic, as steroid use is the most common cause of non-traumatic AVN. The incidence is calculated at 21% without other risk factors and can reach 37% in patients with SLE. Even though the exact duration or dosage is still unidentified, the longer the duration and the higher the dose, the more the risk is increased [3]. The Committee on Nomenclature and Staging of the Association Research Circulation Osseous (ARCO) agreed on the classification criteria of steroid-induced AVN, which are the administration of more than 2 gr of prednisone in a period of over 3 months and the diagnosis being made within 2 years of the steroid use and without any other risk factor [34].

Several factors may contribute to the development of AVN in individuals with a history of COVID-19 infection and steroid therapy. It is important to mention the vascular effects, as COVID-19 has been associated with vascular complications, including thrombosis and damage to blood vessels, but also the disruption of blood supply to the bones that can contribute to AVN. Next, the body’s immune response to the viral infection, as well as the anti-inflammatory effects of steroid therapy, may play a role in disrupting the normal bone repair and maintenance processes. The steroid treatment in itself, particularly at higher doses and for a prolonged duration, can have negative effects on bone health. They may affect bone remodeling and decrease bone density, potentially contributing to AVN. Moreover, some individuals may be more predisposed to developing AVN due to genetic or other factors, as Karpur et al. [32] demonstrated that males are affected more than females, with a ratio of 7/3, and the combination of COVID-19 infection and steroid therapy may exacerbate this susceptibility. Varius treatments exist for COVID-19 depending on the severity of the infection. Corticosteroid and antiviral medication are the most effective so far and commonly used. One of the most important complications of the steroids, linked to the orthopedic world, is the avascular necrosis and usually it is in the femoral head (FHOn), through disrupting the balance between bone formation and resorption, leading to decreased bone density and compromising blood supply to the bones. Veizi et al. [9], in a cohort study of 472 patients, showed an increased percentage of hip AVN in patients who received steroid treatment, as, also, the percentage of joint pain was more than double in this group. However, no relationship was evident between the duration of treatment, cumulative dosage of medication, and presence of osteonecrosis.

The text underscores the importance of healthcare workers’ awareness of potential musculoskeletal complications, especially FHOn, in COVID-19 survivors treated with corticosteroids. Early detection through screening and close follow up is crucial for timely intervention. The incidence of hip AVN in patients recovering from COVID-19 and receiving steroid therapy varies among the studies. Symptoms of hip AVN typically manifested several weeks to months after COVID-19 infection and steroid treatment. The shortest time for the symptoms to develop was illustrated by Kashkosh et al. [15], who published a case report of a 40-year-old male patient with symptoms of left humeral head AVN, 2 days after the second dose of the Pfizer COVID-19 vaccine, on the same shoulder. Next, Sinha et al. [16] in a cohort study of 10 patients, with a mean age of 53.9 years old, who suffered from moderate to severe COVID-19 infection, gathered data of hip AVN at around 14 days later and 4 of them needed core decompression surgery. The majority of articles indicate that the FHOn manifests at 30–60 days after the infection, and in some cases can take up to 8–10 months [17,25].

In our review, 613 patients suffered from moderate to severe SARS infection and 255 of them were not treated with steroids. The following studies highlight the importance of corticosteroid treatment in the development of AVN. Uysal et al. [21] reports a case study of a 63-year-old male patient, who suffered from moderate to severe COVID-19 infection without steroid treatment. Some days later, he had symptoms of hip AVN. Treatment was conservative at first with protected weight bearing and oral alendronate. Due to no improvement, THA followed. Next, Ergun et al. [22] also treated a 51-year-old female patient for COVID-19 infection, without steroids, and 60 days later, she had symptoms of bilateral hip AVN that was improved clinically with core decompression, PT, and no weight bearing for 6 weeks. She did not present femoral head subchondral bone collapse. The 17 patients in Jyothiprasanth et al.’s [23] cohort study, the majority of whom had COVID-19 infection in the past and were not administered any steroid, presented 2 months later with symptoms of hip AVN and 4 of them were bilateral.

A range of steroid doses and treatment durations was observed across the studies. Even in the corticosteroid department there was a big variance, with the scientists using mostly methylprednisolone, prednisolone, and dexamethasone either intravascularly, orally, or in combination. In one study, like the one conducted by Baimukhamedov et al. [10], even the amount of steroids had a wide range, with a cumulative corticosteroid dose at 50–20,675 mg of prednisolone. The biggest portion of steroids was given by Velchov et al. [27], who treated the patients with severe infection by administering 120 mg of dexamethasone with 3600 mg of methylprednisolone in total. A total of 56 days later, the patients had signs of hip AVN and four of them were affected on both hips. All of them except one underwent THA, and five of them had core decompression. Also, Panin et al. [19] medicated the patients with a mean total dose of dexamethasone/prednisolone of 264 (80–600 mg)/1759 (533–4000 mg) and 96.6 days later, they had signs of bilateral hip AVN, which was successfully encountered surgically, with an improvement in motor activity and decrease in pain intensity but no significant improvement in MRI. Regarding the same area, Ardakani et al. [14] administered patients with severe infection a 1695.2 mg mean dose of prednisolone and 40 days later they presented with hip ONC. Generally, the most preferred dosage is around 500 and 800 mg of methylprednisolone [29,30], as Jayapalan et al. [26] presents a young woman with bilateral hip AVN after a combination of oral and IV methylprednisolone and Daltro et al. [35] illustrates 33% osteonecrosis of the femoral head, in patients with severe SARS infection. However, Alkindi et al. [18] combined experimental anti-COVID-19 therapies (favipiravir, hydroxychloroquine, tocilizumab) and 200 mg of methylprednisolone and the patient experienced mild signs of hip AVN, which was improved with NSAIDs and PT.

Another concern after the presentation of the AVN is the course of treatment, which can be nonoperative with NSAIDS, physical therapy [18,22], intra-articular (IA) steroid injections, oral or IV steroid medication, and bisphosphonates [21,24] or operative with core decompression, total hip arthroplasty (THA), and bone marrow aspirate concentrate (BMAC) injection. Concerning the conservative treatment, Sulewski et al. [5], in 70% of patients, after observing no positive effect with NSAIDs and IA injection, proceeded to oral steroids with 10% of them experiencing chronic pain without joint destruction though in the control tests. Also, Alkindi et al. [18] had clinical improvement and Agarwala et al. [24] used oral alendronate and IV zoledronic acid in patients with bilateral hip AVN, which led to an improvement in a mean VAS score of 2.7. As for the surgical intervention, the option that is favored is core decompression often in combination with BMAC and next the THA. It seems that the first type of treatments was chosen mostly for the younger and more active patients [20,30,31] and when THA is chosen, the direct approach is the preferred one [14]. Only one case reported by Ardakani et al. [14] involved infection post-THA due to Serratia marcescens in both hips and needed a two-stage THA with laboratory and clinical improvement postoperatively.

The text has implications for public health, urging healthcare providers to be vigilant in monitoring and managing post-acute complications in COVID-19 survivors, and emphasizes the need for further studies to identify risk factors, determine the incidence of FHOn, and establish protocols for an early diagnosis and intervention.

## 5. Putative Future Guidelines

In order to limit the cases of AVN after COVID-19 infection and steroid treatment, it is important to adapt the cumulative dose of prednisone to a maximum of 2 gr and the duration of the treatment to less than 3 months. Moreover, it is important to reduce as much of the other risk factors for bone ischemia as possible, while also conducting screening tests, with radiographs and MRI of the major joints, for the first 2 years after the infection, in order to detect the signs of bone necrosis early and intervene.

## 6. Conclusions

The association between steroid therapy for COVID-19 and the development of hip AVN is a noteworthy concern even though no relationship is evident between the duration of treatment, cumulative dose of medication, maximum daily dosage received, and presence of AVN. So far, patients with COVID-19 express more pathogenetic factors in developing AVN and especially those treated with corticosteroids. Close monitoring of patients who receive steroids, especially in higher cumulative doses, is crucial for early detection and intervention. Further research with larger cohorts and long-term follow ups is needed, to better understand the causative relationship and optimal management strategies for hip AVN, in the context of COVID-19 and steroid therapy.

## Figures and Tables

**Figure 1 diagnostics-14-00247-f001:**
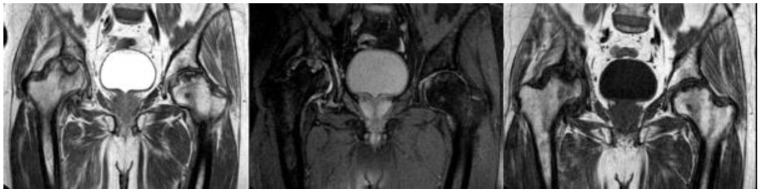
MRI presenting bilateral avascular necrosis of the hip after COVID-19 infection and treatment in ICU unit of our Hospital.

**Figure 2 diagnostics-14-00247-f002:**
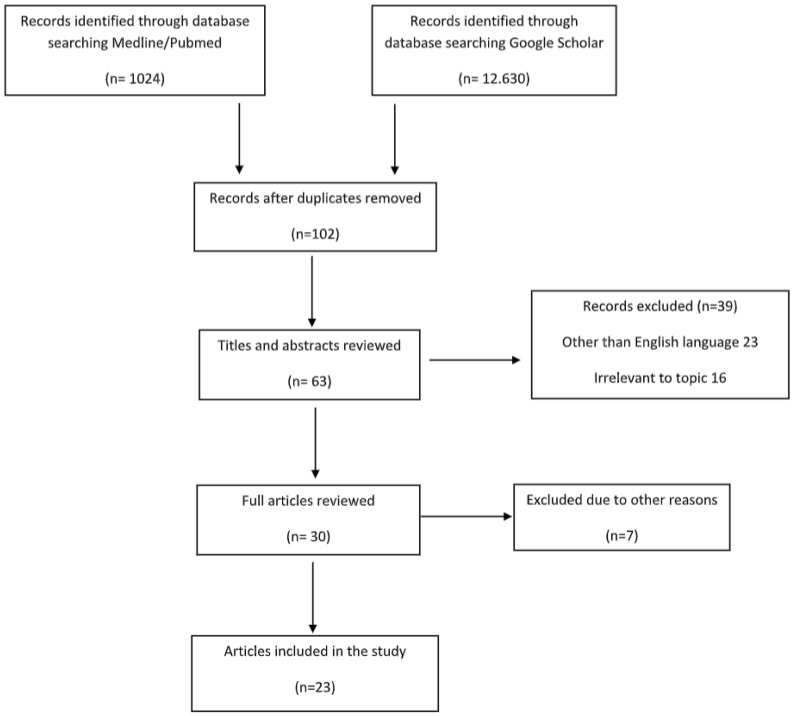
PRISMA Flow Diagram on the study selection process.

**Figure 3 diagnostics-14-00247-f003:**
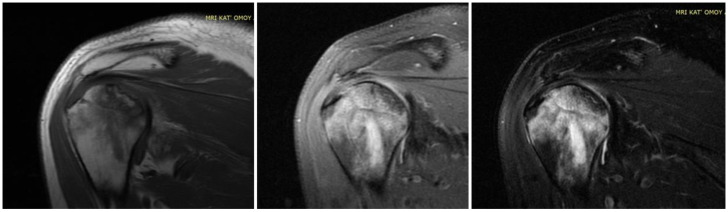
MRI presenting AVN of LEFT humerus head in a case of bilateral osteonecrosis after COVID-19 vaccination and infection treated in ICU with high dose of corticosteroids in our Hospital.

**Figure 4 diagnostics-14-00247-f004:**
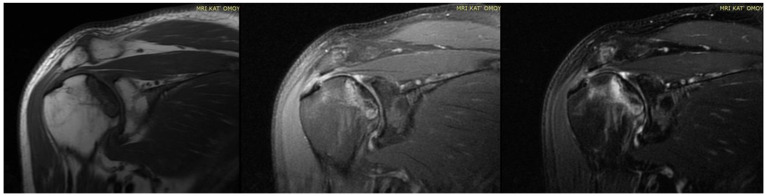
MRI presenting AVN of RIGHT humerus head (previous case of Figure 4) of bilateral osteonecrosis after COVID-19 vaccination and infection treated in ICU with high dose of corticosteroids in our Hospital.

**Figure 5 diagnostics-14-00247-f005:**
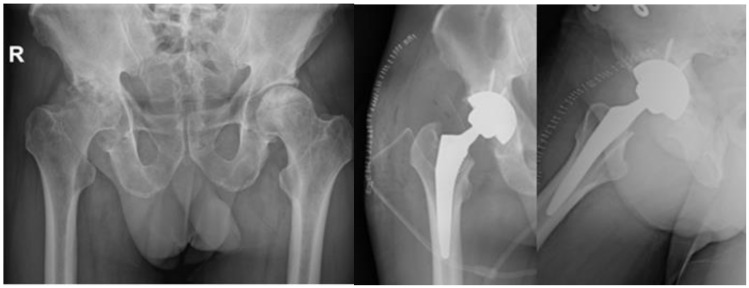
Final treatment of AVN after COVID-19 infection with total hip arthroplasty in our Department.

## Data Availability

Publicly available datasets were analyzed in this study. This data can be found here in https://pubmed.ncbi.nlm.nih.gov/.

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
