# Peer review of "A Comprehensive Review of COVID-19-Infection- and Steroid-Treatment-Associated Bone Avascular Necrosis: A Multi-Study Analysis"

_diagnostics, 2024, doi:10.3390/diagnostics14030247_

Round 1

Reviewer 1 Report

Comments and Suggestions for Authors

The paper proposes a  Comprehensive Review of COVID-19 Infection and Steroid Associated Treatment Bone Avascular Necrosis (AVN).

The materials and methods used are correct and well structured. However, these are only 23 articles with a total of only 613 patients, relatively small numbers compared to the subjects who contracted COVID. The articles considered for review are relevant and appropriate. The discussion is extensive and well organised. 

The images an graphs are clear and correct and the captions appropriate. The bibliography is recent and updated.  

The results of the review are supported and compatible by the data analyzed and collected, but they do not clarify the correlation between COVID, use of steroids and AVN, I believe it is useful that data relating to the frequency of AVN and Steroid in the absence of COVID infection to understand at least how this may have contributed to the onset of this pathology.

I agree with the authors that further research with larger cohorts and long-term follow-up is needed to better understand the causative relationship and optimal management strategies for hip AVN in the context of COVID-19 and steroid therapy .

Author Response

Thank you for your constructive feedback. We have added an additional paragraph in the discussion section, analyzing the impications of steroid treatment in the bone necrosis as you have suggested. 

Reviewer 2 Report

Comments and Suggestions for Authors

The manuscript entitled, 'A comprehensive review of COVID-19 infection and steroid associated treatment bone Avascular necrosis: a multi-study analysis' written by Dr. Evangelia and coauthors is an honest attempt to understand the link between steroid treatment of COVID-19 survivors and the development of ANFH. The paper is well-written and nicely compiled.  Authors are suggested to rectify/edit the following few points for a better understanding of the paper.

1. Recheck the title, it should be, ‘A comprehensive review of COVID-19 infection and steroid associated treatment IN bone avascular necrosis: a multi-study analysis’ rather than, ‘A comprehensive review of COVID-19 infection and steroid associated treatment bone Avascular necrosis: a multi-study analysis’. So put the word ‘in’ in this title at the place suggested.

2. The authors have rightly written that avascular necrosis is related to endothelial dysfunction, but lately, it has been put forth that angiogenesis and coagulopathy also contribute to Avascular necrosis significantly. Therefore, write a few lines regarding these two aspects also in the Introduction section.

3. Citation of references in the text should be like (12,17,22) or (18,19,25) rather than (12)(17)(22)-see lines 132,133. So check the whole text of the manuscript and correct these.

4. Can authors write a paragraph ‘putative future guidelines’ before the conclusions? It will increase its citation capacity

Author Response

Thank you for your constructive feedback. We have adjusted the manuscript title and made the modifications concerning the references numbering. Also we added additional informations in the Introduction section about the disrupted molecular pathways of AVN and the section of putative futere guidelines.